# Modelling a Supplementary Vaccination Program of Rubella Using the 2012–2013 Epidemic Data in Japan

**DOI:** 10.3390/ijerph16081473

**Published:** 2019-04-25

**Authors:** Taishi Kayano, Hyojung Lee, Hiroshi Nishiura

**Affiliations:** 1Graduate School of Medicine, Hokkaido University, Kita 15-Jo Nishi 7-Chome, Kita-ku, Sapporo-shi, Hokkaido 060-8638, Japan; taishi.kaya@med.hokudai.ac.jp (T.K.); hyojunglee@med.hokudai.ac.jp (H.L.); 2CREST, Japan Science and Technology Agency, Honcho 4-1-8, Kawaguchi, Saitama 332-0012, Japan

**Keywords:** Rubella virus, statistical model, supplementary immunization, outbreak, Epidemiology, Japan

## Abstract

From 2012–2013, Japan experienced a major epidemic of rubella, involving a total of 12,614 rubella cases and 45 confirmed cases of congenital rubella syndrome (CRS). One of the contributory factors in this outbreak may have been that the majority of adult males remained unvaccinated. To plan for a supplementary immunization program (SIP) to elevate the herd immunity level, it is critical to determine the required amount of vaccine and identify the target age groups among males for the SIP. The present study aimed to answer these policy questions, employing a mathematical model and analyzing epidemiological datasets from 2012–2013. Our model allowed us to reconstruct the age- and sex-dependent transmission patterns, and the effective reproduction number during the exponential growth phase in 2013 was estimated to be 1.5. The computed next-generation matrix indicated that vaccinating adult males aged from 20–49 years in 2013, using at least 17 million doses, was considered essential to prevent a major epidemic in the future. The proposed model also indicated that, even with smaller doses of vaccine, the SIP in adult males could lead to a substantial reduction in the incidence of rubella, as well as CRS. Importantly, the present study endorses a substantial background risk of observing another major epidemic from 2018–2019, in which cases may be dominated by adult males aged from 25–54 years, that is, our identified age groups plus a five-year time lag from 2013 to 2018.

## 1. Introduction

Rubella, or so-called German measles, is a vaccine-preventable disease caused by rubella virus [1]. Although rubella virus infection mostly results in asymptomatic or mild infections, it can lead to clinically apparent infections in a portion of infected individuals, and the symptoms include a generalized rash (exanthem), lymphadenopathy, joint pain, headache and conjunctivitis [1,2]. While the majority of infections are self-limiting, the infection in pregnant women frequently leads to pregnancy complications, including miscarriage, stillbirth and congenital rubella syndrome (CRS), especially seen in infections during the first trimester of pregnancy [3,4]. The clinical complications of CRS vary, but frequently involve deafness, cataracts and congenital heart disease [1,4,5]. Two distinct strategies have been implemented to prevent rubella in pregnant women, that is, one targeting pregnant women for vaccination, anticipating a direct effect, and the other targeting infants of both sexes, expecting to elevate the level of herd immunity. With the goal of eventual control of rubella at the population level, many countries aim to achieve sufficient herd immunity via a routine mass immunization program [6,7,8].

From 2012–2013, Japan experienced a major epidemic of rubella, while the country was considered on its way to elevating herd immunity through a continued vaccination program and eliminating rubella in the future [9,10]. Although smaller than the outbreaks notified in the 1990s, a total of 12,614 rubella cases, including 9,692 male and 2,922 female cases, were reported to the government, and as many as 45 cases of CRS were confirmed and notified (Figure 1A). A critically important characteristic of this recent outbreak in Japan was that the majority of rubella cases were in males aged in their 20s to 40s, with a smaller portion of females aged in their 20s to 30s infected (Figure 1B). This was attributable to the past vaccination policy in Japan [9]. Rubella-containing vaccine became available in the late 1970s, and up until 1995, the policy of the Japanese government to prevent rubella was aimed at ensuring individual protection among females, routinely vaccinating junior high school girls. As the scientific evidence became available, the government decided to switch their target to infants of both sexes, aged from 12–90 months, with the aim of elevating herd immunity and lowering the frequency of transmission at the community level [11]. However, men who remained unvaccinated were left susceptible and unfortunately became the victims of recent outbreaks, allowing for the persistence of virus transmission in the population [10,12].

As susceptible individuals remain, another epidemic has currently been recognized (2018–2019) [13]. Although the Japanese government set a goal to achieve rubella elimination by 2020, which coincides with the Tokyo Olympic games, allowing continued chains of transmission to occur instills doubt as to the possibility of elimination at this stage. Accordingly, it is evident that the country needs to plan a supplementary immunization program (SIP) to ensure sufficient herd immunity, by means of vaccination of adults [14,15,16]. To plan an effective vaccination strategy using non-excessive doses, it is vital to identify the optimal target host (i.e., age and sex) and calculate the required number of vaccine doses. The purpose of the present study is to clarify the vaccination subjects and the required number of doses for the SIP in Japan via the analysis of epidemiological datasets from 2012–2013.

## 2. Materials and Methods

### 2.1. Epidemiological Data

We analyzed the outbreak data from 2012–2013 to reconstruct the transmission dynamics by age and sex in Japan, and subsequently, considered scenario analysis for the SIP for different age and sex groups. To reconstruct the dynamics, we collected epidemiological data that consisted of weekly notifications of rubella and CRS cases from infectious disease surveillance; both rubella and CRS are classified as category V diseases according to the Infectious Diseases Law in Japan, and all physicians diagnosing these diseases must notify the case to the government via local health centers. The diagnostic criteria for a confirmed case of rubella rest on the detection of IgM antibody by a hemagglutination-inhibition (HI) assay, and at least one of the following symptoms: a generalized rash, fever or lymphadenopathy. The confirmatory diagnosis of CRS is made by Polymerase Chain Reaction (PCR) testing, and at least one of the following symptoms: cataracts, congenital glaucoma, congenital heart disease, deafness (or hearing loss), pigmentary retinopathy, purpura, splenomegaly, microcephaly, mental retardation, meningoencephalitis, a radiolucent lesion in bone, or jaundice within 24 h of birth.

Rubella incidence and demographic datasets of males and females were divided into 5-year age groups, so that the data could be analyzed using a discrete-age mathematical model. That is, our subject ages were grouped as 0–4, 5–9, 10–14, 15–19, 20–24, 25–29, 30–34, 35–39, 40–44, 45–49, 50–54, 55–59, 60–64, 65–69 and ≥70 years old, that is, in total, there were 15 age groups (*n_a_* = 15). In addition to the analysis of rubella case data, we predicted the number of CRS cases using the expected number of rubella cases in adult females [17]. To do so, we analyzed not only rubella case data, but also the vital statistics associated with child delivery. To quantify the age distribution of pregnant women in 2013, we collected the demographic dataset of females and livebirths in 2013 from the Statistics Bureau in Japan [18]. We assumed that mothers’ ages at delivery ranged from 15 to 49 years, that is, there were a total of seven age groups, which were mothers aged from 15–19, 20–24, 25–29, 30–34, 35–39, 40–44 and 45–49 years. Similarly, the number of livebirths was categorized according to mothers’ ages.

### 2.2. Mathematical Models

We employed mathematical models for the three steps of evaluation. First, we reconstructed the transmission dynamics of the rubella outbreak in 2013, so that age- and sex-dependent transmissions were captured by a set of simple equations. Second, we predicted the cumulative number of CRS cases throughout the course of an epidemic. By formulating the CRS model that arises from rubella cases in adult women, it became possible to examine the decrease (or increase) in CRS as a result of the SIP. Third, we considered scenario analysis of the SIP in adults, especially among males. We considered not only rubella transmission in relation to vaccination, but also ensured that the expected number of CRS cases was reduced by the vaccination program [19].

#### 2.2.1. Transmission Model

Our analysis specifically focused on the exponential growth phase of cases in 2013, because this phase showed the steepest growth of cases during the period from 2012–2014 and the results would thus yield the most pessimistic scenario of vaccination in a conservative manner. To capture the transmission dynamics, the following renewal equation was employed:(1)$$j_{ax,t}=\overset{t-1}{\underset{\tau =1}{\sum }}\overset{n_{a}}{\underset{b=1}{\sum }}\underset{y=\left\{M,F\right\}}{\sum }R_{abxy}j_{by,t-\tau }g_{\tau }$$
where $j_{ax,t}$ represents the number of newly infected rubella cases of age group a and sex x in week t, where x={M, F}, males and females are denoted by M and F, respectively, $g_{\tau }$ is the probability mass function of the generation time of rubella, that is, the time interval from infection in a primary case to infection in the secondary case caused by the primary case, assumed to follow an exponential distribution with a mean of 18 days [20,21], $R_{abxy}$ is the reproduction number, interpretable as the so-called next generation matrix defined as the average number of secondary cases in age group a and sex x, produced by a single primary case of age group b and sex y. In the present study, we decomposed $R_{abxy}$ as:(2)$$R_{abxy}=s_{ax}m_{abxy}$$
where $s_{ax}$ represents the relative susceptibility that leads to successful transmission given a contact in age group a and sex x is subject to exposure, and $m_{abxy}$ is the contact rate between a person of age group a and sex x and the another person of age group b and sex y, as extracted from our recent survey [22]. The effective reproduction number, $R_{e}$, that is, the average number of secondary cases generated by a single primary case in the presence of immune individuals, is derived from the largest eigenvalue of the next generation matrix, $\left\{R_{abxy}\right\}$. We assumed that the weekly count of rubella cases $j_{ax,t}$ follows a Poisson distribution, and the likelihood function with unknown parameters $\theta =\left\{s_{ax}\right\}$ was modelled as:(3)$$L\left(\theta ;j_{ax,t}\right)=\underset{t}{\prod }\underset{a}{\prod }\underset{x=\left\{M,F\right\}}{\prod }\frac{E{\left(j_{ax,t}\right)}^{j_{ax,t}}\exp\left(-E\left(j_{ax,t}\right)\right)}{j_{ax,t}!}$$
where E(.) stands for the expected value, as obtained from the right-hand side of the renewal Equation (1). The growth data from week 5 to 15 in 2013 were analyzed.

#### 2.2.2. Prediction of Congenital Rubella Syndrome

The outbreak datasets for the rubella cases in adult women and CRS cases from 2012–2014 were translated into the estimate of the number of CRS cases. The cumulative number of CRS cases, denoted by crs, throughout the course of an epidemic is modeled by accounting for the number of livebirths in females of age group $a^{\prime }$ (i.e., $a^{\prime }$ = {15–19, 20–24, ..., 45–49}) along with rubella cases in women of the same age group $a^{\prime }$ as:(4)$$E\left(crs\right)=k\overset{2014}{\underset{t=2012}{\sum }}\underset{a^{\prime }}{\sum }\frac{B_{a^{\prime },t}}{N_{a^{\prime },t}}c_{a^{\prime },t}$$
where $c_{a^{\prime },t}$ represents the number of rubella cases in women of age group $a^{\prime }$ in year t, $B_{a^{\prime },t}$ is the number of livebirths by mother’s age group $a^{\prime }$ in year t, and $N_{a^{\prime },t}$ is the number of adult women of age group $a^{\prime }$ in year t. The parameter k is a scaling factor that mechanistically encompasses but is not limited to: (i) the probability that a woman is pregnant at a given point in time; (ii) the probability that rubella in the mother occurs during the early gestational period; and (iii) the probability that the virus is successfully delivered from mother to fetus, resulting in CRS. We assume that the cumulative number of CRS follows a Poisson distribution, and the likelihood function is:(5)$$L\left(k;crs\right)=\frac{E{\left(crs\right)}^{crs}\exp\left(-E\left(crs\right)\right)}{crs!}$$

#### 2.2.3. SIP Scenarios

We aimed to identify the optimal age group for supplementary immunization, by which the effective reproduction number could be minimized using the minimum total doses of vaccines. To do so, we calculated the effective reproduction number for different vaccination doses with different subjects of vaccination (i.e., age group and sex). Vaccine doses varied from 5 to 17 million by every 4 million doses (i.e., 5, 9, 13 and 17 million doses), and the subject of the SIP is restricted to males. The width of age groups to receive vaccination can be varied, that is, the total amount of vaccine can be distributed to a certain age group spanning 10 years (e.g., those aged from 20–29 years), 20 years (e.g., those aged from 30–49 years) or 30 years (e.g., those aged from 20–49 or 30–59 years). With these settings, we numerically explored the effective reproduction number of all possible combinations of target hosts and identified the target group with global minimum value. Assuming that random vaccination takes place within the same age group with coverage $p_{x}$, the effective reproduction number under the SIP $R_{e}$ is calculated as the largest eigenvalue of the next generation matrix {$k_{abxy}$}, which reflects the reduction in the number of secondary transmissions due to vaccination, that is, $k_{abxy}=\left(1-p_{x}\right)R_{abxy}$ for the elements in each entire row of the vaccinated subjects.

Subsequently, we calculated the cumulative number of rubella cases using the final size equation:(6)$$z_{ax}=1-exp\left(-\overset{n_{a}}{\underset{b=1}{\sum }}\underset{y=\left\{M,F\right\}}{\sum }R_{abxy}z_{by}\right)$$
where $z_{ax}$ represents the final size of rubella cases of age a and sex x. Note that the recursive Equation (6) does not have an analytical solution, but the equation can be iteratively solved to identify the final size $z_{ax}$ under an SIP. To quantify the effectiveness of the SIP, the cumulative number of rubella cases, as well as CRS cases, under various vaccination scenarios were computed and compared.

The mean generation time of rubella, that is, the time from infection in the primary case to infection in a secondary case directly transmitted by the primary case, involves a certain extent of uncertainty, as it rests on published historical data [21]. To partly address this uncertainty, a sensitivity analysis of the minimum required dose was conducted for the effective reproduction number to lower the value of one. The mean generation time varied from 9 to 27 days, which is plus/minus 50% of the published mean, 18 days.

### 2.3. Ethical Considerations

The present study used publicly available data that are available elsewhere [23]. The datasets had already been fully anonymized and did not include any identity information. Thus, ethical approval was not required for the analysis.

### 2.4. Data Sharing Policy

Rubella and CRS data are accessible [23].

## 3. Results

The estimated relative susceptibilities, $s_{ax}$, by age and sex are shown in Table 1, with the highest estimate seen in males aged 35–39 years. With the exception of the older age groups, the estimate in each age group was higher in males than in females. Moreover, a relatively higher value than other age groups was identified among males aged 20–44 years, which was consistent with the seroprevalence data [9]. Multiplying the estimated $s_{ax}$ to the known age- and sex-dependent contact matrix $m_{abxy}$, the effective reproduction number ($R_{e}$) was estimated to be 1.53 (95% confidence intervals (CI): 1.48, 1.58) during the exponential growth phase in 2013. That is, on average one primary case produced 1.5 secondary cases during the early stage of the 2013 epidemic.

Figure 2 compares observed and predicted rubella cases during the exponential growth phase in 2013. Qualitatively, the observed patterns were well-captured, even though our model was kept simple. That is, using the contact matrix that is weighted by the relative susceptibility, the differential incidence pattern by age and sex was reconstructed.

As the next step, we explored various possible scenarios in which the SIP was implemented in advance of the epidemic. Allocating different amounts of vaccine to different age groups, we examined how $R_{e}$ varied (Figure 3). Only by using up to 13 million doses for the SIP, was it impossible to lower $R_{e}$ below the value of 1. Nevertheless, we identified males aged from 20–49 years as the optimal group for vaccination, minimizing $R_{e}$ below the value of 1 using 17 million doses (Figure 3A). The examined doses 5, 9, 13 and 17 million correspond to the coverages 20.6%, 37.1%, 53.6% and 70.1%, respectively. Figure 3B,C show the cumulative number of rubella cases for males and females, respectively, that are expected to be averted by an optimal SIP with 5, 9 and 13 million doses, vaccinating males aged from 20–49 years. Even though $R_{e}<1$ is not achieved with these doses, a dramatic decline in the number of cases was expected. Especially, when substantial doses were secured, it is noteworthy that female incidence was also greatly reduced due to an indirect effect. For instance, among working and child-bearing age adults of 20–49 years, 5,279 and 1,166 cases in males and females were anticipated, respectively, with 5 million doses, but only 373 cases in males and 73 cases in females were expected with 13 million doses.

The ratio of the expected number of CRS cases under the SIP to that without the SIP is shown in Figure 4. To calculate this, we used the estimated scaling parameter k at 0.45 (95% CI: 0.28, 0.64). Without the SIP, the expected number of CRS cases was 45.0 (95% CI: 27.8, 63.4), which coincided with the observed cumulative count of CRS cases (i.e., n = 45). Under the SIP scenarios in males aged from 20–49 years, the ratio decreased to 0.75 with 5 million and 0.04 with 13 million doses. With 17 million doses, CRS would not be expected as the major epidemic itself is expected to be prevented in our model. Vaccinating female groups of reproductive age did not yield a comparable reduction in the number of CRS cases, indicating that elevating herd immunity by vaccination of adult males would be more effective.

Figure 5 examined the sensitivity of the minimally required doses of vaccines for the SIP to various values of the mean generation time. While all of our above-mentioned results rested on the assumed mean generation time of 18 days, smaller doses might be enough to prevent a major epidemic if the generation time is shorter. However, if the target age group is different from 20–49-year-olds and the generation time is longer, far greater doses than 17 million would be required to prevent the epidemic.

## 4. Discussion

We retrospectively examined the 2012–2013 epidemic of rubella in Japan, using an epidemiological model. We were motivated by the need to analyze 2012–2013 data, because there is documented evidence that no substantial elevation of immune fraction was observed following this epidemic [24]. The exponential growth phase in 2013 was successfully reconstructed, capturing the age- and sex-dependent transmission patterns, and the effective reproduction number during the corresponding period was estimated to be 1.5, which is consistent with the notion that the country has been on its way to achieving substantial herd immunity [9]. However, relative susceptibility among adult males was estimated to be high, reflecting the fact that substantial numbers of susceptible hosts remain in those cohorts. Thereby, to prevent a major epidemic, 17 million doses of vaccine would be needed, focusing on male subjects aged from 20–49 years, for a SIP. As 17 million is a large number of doses, we also examined how the cumulative incidence is lowered by a SIP with a smaller amount of vaccine doses, showing that the incidence both in males and females, as well as CRS cases, were decreased.

To our knowledge, the present study is the first to explicitly examine the required vaccine doses for a SIP in Japan using the 2012–2013 data. Using the age- and sex-dependent contact matrix in Japan [22], the next generation matrix was fully quantified, allowing us not only to capture the transmission dynamics but also explore the epidemiological effectiveness of various SIP scenarios [25,26]. As a consequence, we have identified that planning a SIP in males aged from 20–49 years would be the most effective at reducing the reproduction number as well as the incidence, although the required dose was indicated to be as large as 17 million. In present day Japan, that is, 2019, males aged from 40 to 57 years missed the opportunity to be vaccinated and those aged 29 to 39 years had the chance of a single dose only. Those aged from 29–57 years in 2019 are roughly captured by our suggestion of 20–49-year-olds in 2013 (i.e., 26–55 years in 2019 accounting for a constant lag of 6 years), and it is remarkable that only 5-year age-shift of susceptible fraction with an assumed stable age- and sex-dependent contact matrix can explain the future observed age-dependent patterns of cases from 2018–19 (the relevant findings are our ongoing study). Our modelling exercise excellently endorsed the notion that those who missed vaccination or had only a single shot need to be vaccinated to prevent another major epidemic in Japan. With substantial doses, we could anticipate a population-level benefit, as observed from similar catch-up campaigns, playing a significant role in the elimination of rubella and CRS in the Americas [27]. Even if the coverage of the SIP was not enough to prevent a future epidemic, it should be noted that a substantial reduction in the incidence is anticipated by vaccinating males in those age groups.

In addition to modelling the epidemiological dynamics of rubella, we have also devised a model to anticipate CRS from rubella in pregnant mothers. To do this, we used rubella data in adult females, demographic data of livebirths and the distribution of mothers’ ages at delivery, and a precise version of the predictive model can be found elsewhere [17]. In the present study, we used the CRS prediction model to anticipate the cumulative incidence of CRS under various SIP scenarios, and the quantified model predicted the same count (i.e., 45 cases) of CRS from 2012–2014. Considering that rubella is a self-limiting disease, anticipating CRS would be critical to evaluate the public health benefit of a SIP. In particular, we examined whether to focus on males aged from 20–49 years or adult females expecting to be pregnant; the former intends to elevate herd immunity, while the latter aims for individual-based protection from severe outcomes. We have shown that vaccinating females did not help to reduce both the incidence of rubella and CRS. Thus, vaccinating females is not theoretically supported even during the course of an epidemic, and given substantial doses, it would be preferable to vaccinate susceptible adult males as much as possible to reduce both rubella and CRS.

While the generation time of rubella was assumed to be 18 days [20,21,22,23], the fixed parameter involved a certain extent of uncertainties, and thus, we examined the sensitivity of our modelling results to different mean generation times [21,28,29]. Our results indicated that the required doses of vaccine can be inflated by longer mean generation times. This is because our estimation was based on the exponential growth phase in 2013, and the mean generation time played a role in translating the growth rate in multiple age- and sex-groups into the reproduction number. Thus, while the abovementioned model emphasized 17 million as the number of possible target doses, it must be remembered that the required dose can increase with a longer mean generation time.

Several limitations must be acknowledged. First, our scenario analysis rests on the next generation matrix quantified using the early growth phase data in 2013, and considering that the corresponding growth phase involved a steep increase in the incidence, our results may represent a pessimistic scenario (i.e., $R_{e}$ could have been smaller). The relative susceptibility in both males and females might have been overestimated. Second, while the use of seroepidemiological data is adopted in several other modelling studies [30,31,32], seroprevalence data were not considered because it was hard to assume that seropositive individuals were fully protected and because we adopted so-called “leaky” vaccine protection in our model. However, even with the simplistic leaky assumption, we were able to show that relative susceptibility was high among adult males aged from 25–39 years. Third, we did not consider spatial data, and it is known that the most intense geographic areas of transmission are urban locations [33]. For example, the predicted incidence of CRS may be greater than we estimated, as there is a higher number of women of child-bearing age are in urban areas than in rural locations. Fourth, exposure to rubella during the early gestational stage can cause not only CRS but also unwanted pregnancy outcomes including miscarriage and stillbirths [3,34]. Because our study relied on the observed data, accounting for these aspects requires us to investigate the natural history further. Fifth, in addition to the required dose for SIP, it must be noted that the target group for vaccination would be mainly adult males [35]. Not only dose calculation, but also the practical implementation of SIP requires the idea to reach to those adults, communicate the risk of rubella appropriately, and motivate them to undertake vaccination (e.g., via involvement of occupational physicians at working place).

Despite these limitations, the present study successfully identified target age groups for a SIP in males and calculating the required doses of vaccine for prevention. Our findings will hopefully encourage medical workers, those in the public health sectors and policy-makers who address the elimination of rubella and CRS, to take significant steps to recognize the impact of an SIP.

## 5. Conclusions

We retrospectively examined the 2012–2013 epidemic of rubella in Japan, using an epidemiological model. The effective reproduction number in 2013 was estimated to be 1.5, and an age-and sex-dependent next generation matrix was quantified. To prevent a major epidemic, a SIP needs to be implemented involving the administration of 17 million doses of vaccine, focusing on males aged from 20–49 years. Even with smaller doses, the incidence of rubella and CRS both in males and females would be expected to decrease. Our modelling clearly indicated that adult males who had missed vaccination or had only received a single shot, need to be vaccinated to prevent another major epidemic in Japan.

## Figures and Tables

**Figure 1 ijerph-16-01473-f001:**
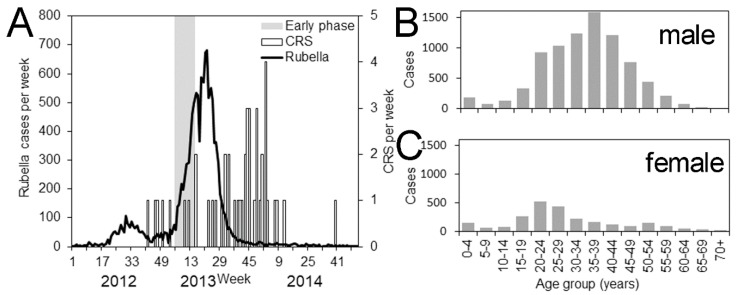
Epidemiological characteristics of the rubella epidemic from 2012–2013 in Japan. (**A**) Epidemic curve of rubella cases (bold line) and notification of congenital rubella syndrome (bars). Epidemic curve is the weekly surveillance dataset of rubella that rests on notifications adhering to the Infectious Disease Law of Japan. The gray shaded area represents approximately a linear growth period in 2013 from which we quantified the age- and sex-dependent next generation matrix. (**B**,**C**) Age distribution of notified rubella cases for males and females, respectively.

**Figure 2 ijerph-16-01473-f002:**
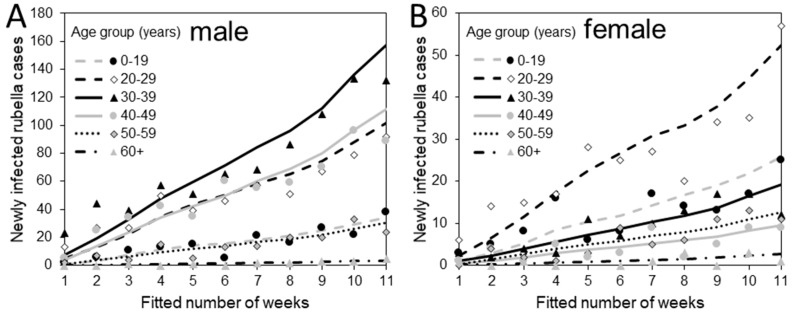
Comparisons between the observed and predicted weekly incidence of rubella by age and sex in Japan, 2013. (**A**) Comparisons between the observed and predicted weekly incidence of males and (**B**) females. Lines represent predictions that were derived from our multivariate renewal process model. Marks represent the observed data by age group.

**Figure 3 ijerph-16-01473-f003:**
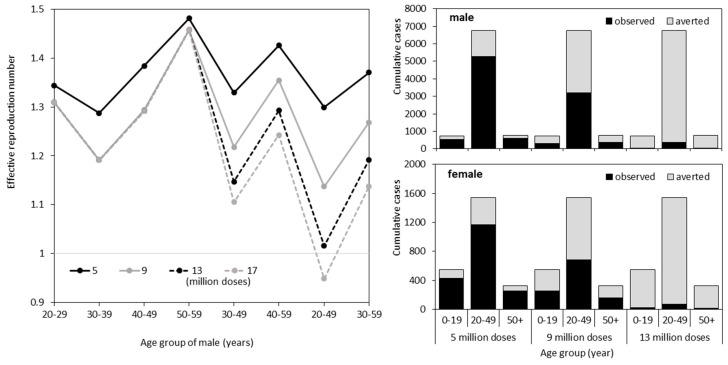
Supplementary immunization programs (SIPs) among different age groups. (**A**) Effective reproduction numbers under different SIP scenarios, varying doses and subjects. Resulting estimates using an identical dose are connected by lines. A gray horizontal axis shows a threshold value (=1) below which a major epidemic is prevented. (**B**,**C**) The cumulative numbers of rubella cases associated with different SIP are shown for each age group for males and females, respectively. Assuming that the SIP was conducted in males aged from 20–49 years, the cumulative number of cases was calculated. Black bars represent the expected number of cases under different SIP scenarios, and gray bars represent cases averted by the SIP.

**Figure 4 ijerph-16-01473-f004:**
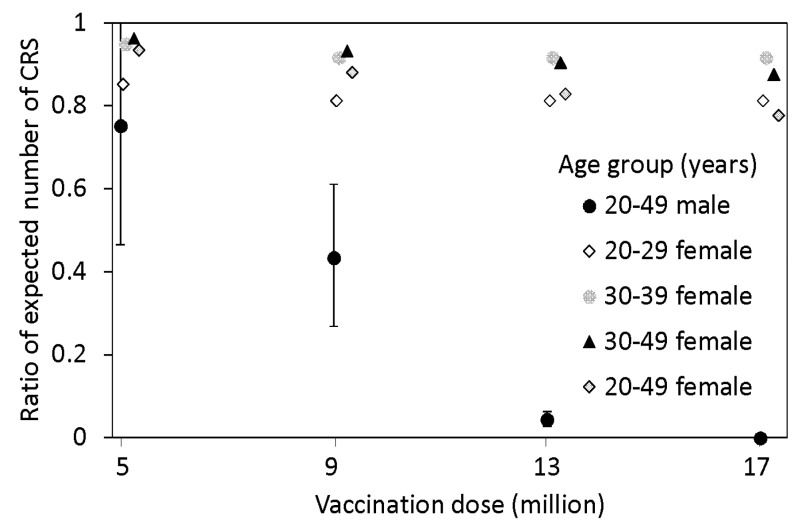
Ratio of the expected number of congenital rubella syndrome (CRS) cases under the SIP to that without the SIP. Dots represent the ratios under different SIP scenarios with an identical subject for vaccination. Whiskers describe the 95% confidence intervals of each ratio, for males only.

**Figure 5 ijerph-16-01473-f005:**
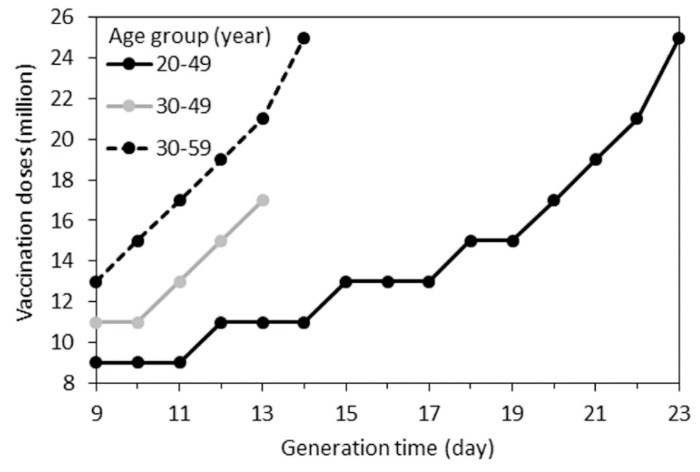
Sensitivity of the required minimum doses of vaccine to the mean generation time. SIP scenarios with an identical subject for vaccination are connected by lines. The minimum required doses are shown from 9 to 23 days of the mean generation time of rubella. Although we investigated mean generation times that varied from 9 to 27 days, the specified range from 9 to 23 days is shown, because $R_{e}$ was not expected to decrease to less than one by vaccinating the corresponding group. more.

**Table 1 ijerph-16-01473-t001:** Relative susceptibility, interpreted as being proportional to the per contact probability of transmission, depending on age group and sex.

Age Group	Male	(95% CI ^a^)	Female	(95% CI)
0–4	0.286	(0.207, 0.392)	0.081	(0.052, 0.127)
5–9	0.085	(0.055, 0.132)	0.058	(0.036, 0.092)
10–14	0.134	(0.098, 0.183)	0.082	(0.056, 0.121)
15–19	0.196	(0.158, 0.242)	0.185	(0.148, 0.232)
20–24	0.419	(0.371, 0.473)	0.292	(0.249, 0.341)
25–29	0.441	(0.393, 0.494)	0.354	(0.298, 0.420)
30–34	0.524	(0.473, 0.580)	0.145	(0.111, 0.190)
35–39	0.606	(0.553, 0.664)	0.104	(0.078, 0.138)
40–44	0.486	(0.439, 0.538)	0.055	(0.038, 0.080)
45–49	0.297	(0.259, 0.340)	0.054	(0.036, 0.081)
50–54	0.170	(0.140, 0.207)	0.104	(0.077, 0.140)
55–59	0.116	(0.090, 0.149)	0.075	(0.051, 0.112)
60–64	0.030	(0.017, 0.054)	0.028	(0.013, 0.059)
65–69	0.023	(0.010, 0.050)	0.034	(0.015, 0.079)
70+	0.004	(0.001, 0.026)	0.005	(0.001, 0.025)

^a^ CI: confidence intervals.

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
