# Peer review of "Modelling a Supplementary Vaccination Program of Rubella Using the 2012–2013 Epidemic Data in Japan"

_ijerph, 2019, doi:10.3390/ijerph16081473_

Round 1

Reviewer 1 Report

The manuscript entitled "Modelling a supplementary vaccination program of rubella using the 2012–2013 epidemic data in Japan" is a very interesting approach to use historical data on a highly contagious disease epidemic to model the impact of different immunization strategies. The paper is very well written and uses a very clear argumentation. Since I'm not an expert modeller, the formulas could be double checked by a mathematician.

I have only one concern that the epidemiologic situation in 2012/2013 does not necessarily have to predict unambiguously the situation in 2018/2019. First, more birth cohorts were vaccinated that were playing the major role in disease spread in the past. Because the contact patterns differ by age, the epidemic could express different dynamics. The use of specific Japanese contact pattern study is comforting and maybe the authors consider this shift of transmission age in their model? Second, many of the young males which were the main reservoir for the 2012/2013 epidemic could have acquired immunity in 2013 by subclinical infection or boosting the immunity if vaccinated with the first dose. Therefore it is a bit surprising that the authors have ignored the seroprevalence data which could enable some insight into this aspect of susceptibility of the particular birth cohorts. I think that the model is still a very useful approach to plan supplementary immunization (very challenging in adult populations, as not as easy to outreach like schoolchildren for example). Maybe the authors should emphasize more this limitation by explaining their choices and possible implications in the introduction of the model and then referring to this in the limitations (like they correctly did).

My second (minor) suggestion is about not using only absolute number of doses for particular scenarios. Readers not aware of the Japanese population structure can be puzzled by these numbers (5, 9, 13, 17 million doses). The authors at some stage (maybe in the Methods) could explain to what vaccine uptake in the target groups would these numbers correspond.

Author Response

[Point by point responses to Reviewers: [IJERPH] Manuscript ID: ijerph-479432]

Responses to the Reviewer 1

Reviewer 1: The manuscript entitled "Modelling a supplementary vaccination program of rubella using the 2012–2013 epidemic data in Japan" is a very interesting approach to use historical data on a highly contagious disease epidemic to model the impact of different immunization strategies. The paper is very well written and uses a very clear argumentation. Since I'm not an expert modeller, the formulas could be double checked by a mathematician. I have only one concern that the epidemiologic situation in 2012/2013 does not necessarily have to predict unambiguously the situation in 2018/2019. First, more birth cohorts were vaccinated that were playing the major role in disease spread in the past. Because the contact patterns differ by age, the epidemic could express different dynamics. The use of specific Japanese contact pattern study is comforting and maybe the authors consider this shift of transmission age in their model? Second, many of the young males which were the main reservoir for the 2012/2013 epidemic could have acquired immunity in 2013 by subclinical infection or boosting the immunity if vaccinated with the first dose. Therefore it is a bit surprising that the authors have ignored the seroprevalence data which could enable some insight into this aspect of susceptibility of the particular birth cohorts.

>> 

We thank the reviewer for this very encouraging comment. We have noted that only the 5-year age-shift of susceptible fraction can excellently explain the ongoing epidemic pattern of rubella from 2018-19 in P8L232-235. Moreover, we have a foregoing study by Nishiura and others (2015 IJID) that indicated that there was no substantial elevation of immune fraction positive against rubella following the 2013-14 epidemic (P8L251-253).

I think that the model is still a very useful approach to plan supplementary immunization (very challenging in adult populations, as not as easy to outreach like schoolchildren for example). Maybe the authors should emphasize more this limitation by explaining their choices and possible implications in the introduction of the model and then referring to this in the limitations (like they correctly did).

>> 

We have added the description that it would be difficult to reach to adult male in Discussion (P9L318-322).

My second (minor) suggestion is about not using only absolute number of doses for particular scenarios. Readers not aware of the Japanese population structure can be puzzled by these numbers (5, 9, 13, 17 million doses). The authors at some stage (maybe in the Methods) could explain to what vaccine uptake in the target groups would these numbers correspond.

>> 

We have added the description of coverage, if all were equally distributed to the optimal target group, i.e., adult male aged from 20-49 years in P6L208-209.

Reviewer 2 Report

Thank you for the opportunity to review this important manuscript. The manuscript describes an application of mathematical modeling to address an important public health question in Japan, essentially, to reach the Japanese elimination goal, how large of a nationwide supplementary immunization program (SIP) is needed. The authors describe that the model is based upon the surveillance data collected during the 2011-2013 outbreak, focusing on data from early 2013, which identifies the greatest increase in cases, which would produce the most conservative estimates for designing an SIP. In addition, the authors used the model to predict the number of CRS cases would be expected from future outbreaks. The manuscript is very well written, and quite clear. The authors have spent a considerable amount of time ensuring that it is highly polished. Upon reading the manuscript there are three concerns that need to be addressed, and a few minor comments for consideration. One concern with the paper is the use of identification of the term of the number of vaccine doses needed. While it is recognized that governments need to know the number of doses to procure, in this manuscript how to convert the number of doses to people vaccinated is unclear. It would be helpful to know the estimated coverage of the SIP to achieve the goal. Also, it would be good to estimate the wastage rate expected to estimate ‘number of doses’ needed, if this was included in the figure? The second concern that I had is with the identification of appropriate age group. The text and (Lines 204-216) and figure 3 demonstrate why the 20-49 age group was identified, it is still unclear to me. It is clear that Re is

Author Response

Responses to the Reviewer 2

Reviewer 2: Comments and Suggestions for Authors. Thank you for the opportunity to review this important manuscript. The manuscript describes an application of mathematical modeling to address an important public health question in Japan, essentially, to reach the Japanese elimination goal, how large of a nationwide supplementary immunization program (SIP) is needed. The authors describe that the model is based upon the surveillance data collected during the 2011-2013 outbreak, focusing on data from early 2013, which identifies the greatest increase in cases, which would produce the most conservative estimates for designing an SIP. In addition, the authors used the model to predict the number of CRS cases would be expected from future outbreaks. The manuscript is very well written, and quite clear. The authors have spent a considerable amount of time ensuring that it is highly polished. Upon reading the manuscript there are three concerns that need to be addressed, and a few minor comments for consideration. One concern with the paper is the use of identification of the term of the number of vaccine doses needed. While it is recognized that governments need to know the number of doses to procure, in this manuscript how to convert the number of doses to people vaccinated is unclear. It would be helpful to know the estimated coverage of the SIP to achieve the goal. Also, it would be good to estimate the wastage rate expected to estimate ‘number of doses’ needed, if this was included in the figure?

>> 

We thank the reviewer for these encouraging comments. We have added the description of coverage, if all were equally distributed to the optimal target group, i.e., adult male aged from 20-49 years in P6L208-209.

The second concern that I had is with the identification of appropriate age group. The text and (Lines 204-216) and figure 3 demonstrate why the 20-49 age group was identified, it is still unclear to me. It is clear that Re is

>> 

In P4L158-160, we have clearly indicated that we numerically explored all possible combinations of male host to undertake vaccination.